# MAKING CONVOLUTIONAL NETWORKS SHIFT-INVARIANT AGAIN

## ABSTRACT

Modern convolutional networks are not shift-invariant, despite their convolutional nature: small shifts in the input can cause drastic changes in the internal feature maps and output. In this paper, we isolate the cause – the downsampling operation in convolutional and pooling layers – and apply the appropriate signal processing fix – low-pass filtering before downsampling. This simple architectural modification boosts the shift-equivariance of the internal representations and consequently, shift-invariance of the output. Importantly, this is achieved while maintaining downstream classification performance. In addition, incorporating the inductive bias of shift-invariance largely removes the need for shift-based data augmentation. Lastly, we observe that the modification induces spatially-smoother learned convolutional kernels. Our results suggest that this classical signal processing technique has a place in modern deep networks.

## 1 INTRODUCTION

Deep convolutional neural networks (CNNs) are designed to perform high-level tasks and be robust to low-level nuisance factors. For example, small shifts in the input *should* simply shift the internal feature maps (shift-equivariance), and leave the output relatively unaffected (shift-invariance). This property has been explicitly engineered through convolutional and pooling layers, where the same function is applied on a local region across the image in a sliding window fashion. However, recent work (Engstrom et al., 2017; Azulay & Weiss, 2018) has found that small shifts can drastically change the output of a classification network. Why is this the case?

Shift-invariance is lost when spatial resolution is lost, for example, from pooling layers. Our insight is that conventional strided-pooling, as shown in Fig. 1 (top), is inherently composed of two operations: (1) evaluating the pooling operator densely (without striding), and (2) downsampling. Naive downsampling loses shift-equivariance, as high-frequency components of the signal alias into low-frequencies. This phenomenon is commonly illustrated in movies, where wheels appear to spin backwards, due to the frame rate not meeting the Nyquist sampling criterion (known as the Stroboscopic effect). Separating these operations is important, as it allows us to keep the pooling operation, while applying the appropriate fix to the downsampling operation.

We propose to add the signal processing tool of *low-pass filtering before downsampling*, as shown in Fig. 1 (bottom). By low-pass filtering, the high-frequency components of the signal are reduced, reducing aliasing and better preserving shift-equivariance. This ultimately cascades into better shift-invariance in the output. We show example classification instabilities in Fig. 2.

A potential concern is that over-aggressive low-pass filtering can result in heavy loss of information. However, we find that with a reasonable selection of low-pass filter weights, we can maintain classification performance while increasing shift-invariance. Furthermore, we show that without shift-based data augmentation, incorporating this inductive bias actually improves performance.

We find that the learned filters also naturally become smoother after adding the blurring layer. These results indicate that incorporating this small modification not only induces shift-invariance, but causes the network to learn a smoother feature extractor.

In summary, our contributions are as follows:

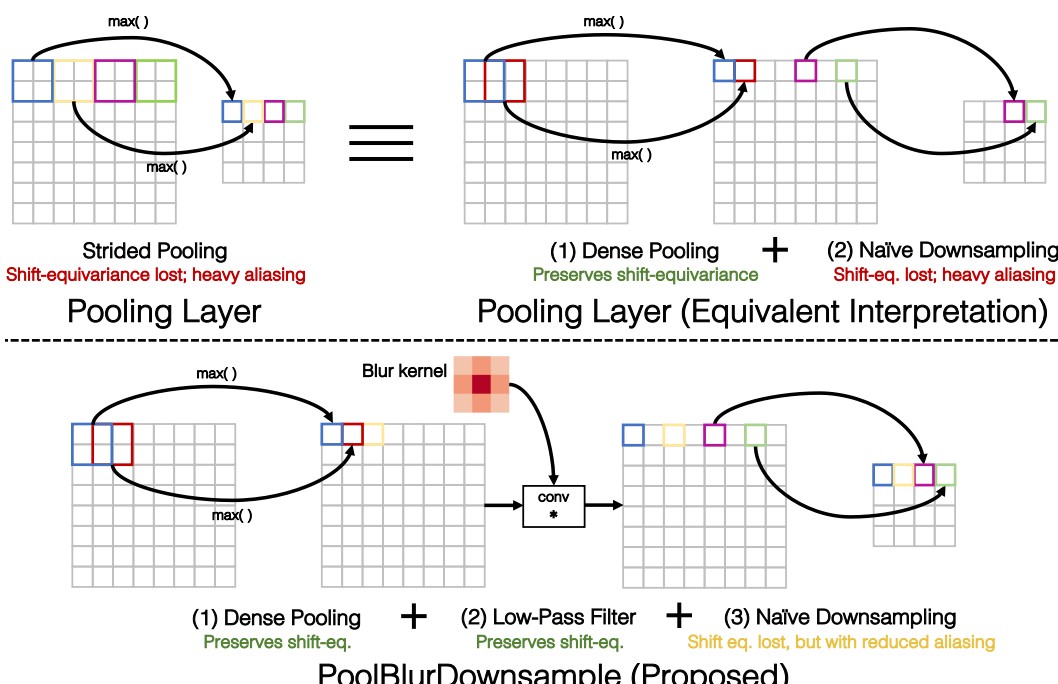

**Figure 1:** **(Top)** Pooling does not preserve shift-equivariance. It is functionally equivalent to densely evaluated pooling followed by naive downsampling. The latter operation ignores the Nyquist sampling theorem and loses shift-equivariance. **(Bottom)** We low-pass filter between the operations. This keeps the original pooling operation, while antialiasing the appropriate signal. This equivalent analysis and modification can be applied to any strided layer, such as convolution.

- We isolate the cause for loss of shift-invariance – downsampling. Separating the downsampling from pooling enables us to keep the desired pooling, while fixing the loss of shift-equivariance. We propose to low-pass filter before downsampling, a common signal processing technique.
- We validate on a classification task, and demonstrate increased shift-equivariance in the features and shift-invariance in the output.
- In addition, we observe large improvements in classification performance when training without shift-augmentation, indicating more efficient usage of data.

## 2 RELATED WORK

Local connectivity and weight sharing have been a central tenet of neural networks, including the Neocognitron (Fukushima & Miyake, 1982), LeNet (LeCun et al., 1998) and modern networks such as Alexnet (Krizhevsky et al., 2012), VGG (Simonyan & Zisserman, 2014), ResNet (He et al., 2016), and DenseNet (Huang et al., 2017). In biological systems, local connectivity was famously discovered observed in a cat's visual system by Hubel & Wiesel (1962). Recent work has strived to build in additional types of invariances, such as rotation, reflection, and scaling (Sifre & Mallat, 2013; Bruna & Mallat, 2013; Esteves et al., 2017; Kanazawa et al., 2014; Worrall et al., 2017; Cohen & Welling, 2016). Our work focusses on the elusive goal of shift-invariance.

Though properties such as shift-equivariance have been engineered into networks, what factors and invariances does an emergent representation actually learn? Analysis of deep networks have included qualitative approaches, such as showing patches which activate hidden units (Girshick et al., 2014; Zhou et al., 2014), actively maximizing hidden units (Mordvintsev et al., 2015), and mapping features back into pixel space (Dosovitskiy & Brox, 2016a;b; Mahendran & Vedaldi, 2015; Zeiler & Fergus, 2014; Nguyen et al., 2017; Hénaff & Simoncelli, 2015). Our analysis is focused on a specific, low-level property and is complementary to these qualitative approaches.

A more quantitative approach for analyzing networks is measuring representation or output changes (or robustness to changes) in response to manually generated perturbations to the input, such as image transformations (Goodfellow et al., 2009; Lenc & Vedaldi, 2015; Azulay & Weiss, 2018),

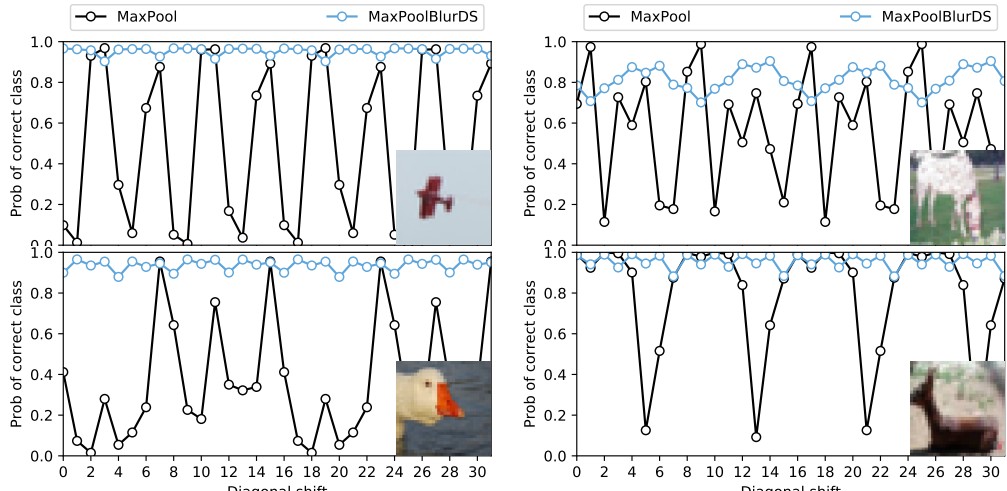

**Figure 2: Classification stability for selected images.** Predicted probability of the correct class changes when shifting the image. The baseline (black) exhibits chaotic behavior, which is stabilized by our method (blue).

geometric transforms (Ruderman et al., 2018; Fawzi & Frossard, 2015), and CG renderings with various shape, poses, and colors (Aubry & Russell, 2015). A related line of work is in adversarial examples, where directed perturbations in the input can result in large changes in the output. These perturbations can be directly on pixels (Goodfellow et al., 2014a;b), a single pixel (Su et al., 2017), small deformations (Xiao et al., 2018), or even affine transformations (Engstrom et al., 2017). We aim make the network robust to the simplest of these types of attacks and perturbations: shifts. Both Hénaff & Simoncelli (2015) and Azulay & Weiss (2018) identify that modern deep networks ignore the Nyquist sampling criterion when downsampling. In our work, we propose and empirically validate an easily adoptable fix which minimally perturbs the existing network architecture.

Classic hand-engineered computer vision and image processing representations, such as SIFT (Lowe, 1999), wavelets, and image pyramids (Burt & Adelson, 1987; Adelson et al., 1984) also extract features in a sliding window manner, often with some subsampling factor. As discussed in Simoncelli et al. (1992), literal shift-equivariance cannot hold when with subsampling. Shift-equivariance can be recovered if features are extracted densely, for example textons (Leung & Malik, 2001), the Stationary Wavelet Transform (Fowler, 2005), and DenseSIFT (Vedaldi & Fulkerson, 2010). Deep networks can also be evaluated densely, by removing striding and making appropriate changes to subsequent layers by using *á trous*/dilated convolutions (Chen et al., 2014; 2018; Yu & Koltun, 2015). This comes at great computation and memory cost. Our work investigates achieving shift-equivariance with minimal additional computation, by blurring before subsampling.

Blurring before downsampling is fundamental technique in signal processing (Oppenheim et al., 1999), image processing (Gonzalez & Woods, 1992), computer graphics (Foley et al., 1995), and vision (Szeliski, 2010). In deep learning, average pooling (LeCun et al., 1990) is a form of blurring. Scherer et al. (2010) finds max-pooling to more effective than variants of blurred-downsampling, under the assumption that they are alternatives. Conversely, we show that they are compatible.

## 3 METHODS

### 3.1 PRELIMINARIES

**Deep convolutional networks as feature extractors** Let an image with resolution $H \times W$ be represented by $X \in \mathbb{R}^{H \times W \times 3}$. An $L$-layer CNN can be expressed as a feature extractor $\mathcal{F}_l(X) \in \mathbb{R}^{H_l \times W_l \times C_l}$, with layer $l \in [0, L]$, spatial resolution $H_l \times W_l$ and $C_l$ channels. Each feature map can also be upsampled to original resolution, $\widetilde{\mathcal{F}}_l(X) \in \mathbb{R}^{H_l \times W_l \times C_l}$.

**Shift-equivariance and shift-invariance** A representation $\widetilde{\mathcal{F}}$ is shift-equivariant if shifting the input produces a shifted feature map, meaning that shifting and feature extraction are commutable. We more rigorously define the Shift function in Eqn. 4.

$$\text{Shift}_{\Delta h, \Delta w}(\widetilde{\mathcal{F}}(X)) = \widetilde{\mathcal{F}}(\text{Shift}_{\Delta h, \Delta w}(X)) \quad \forall (\Delta h, \Delta w) \tag{1}$$

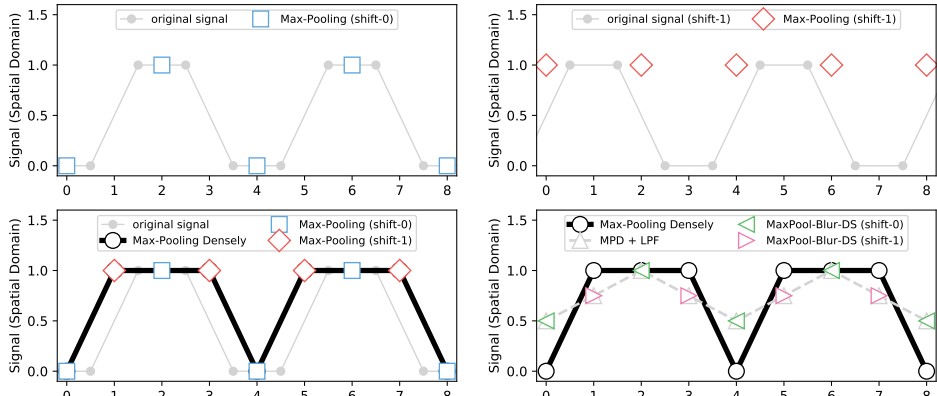

**Figure 3: Toy example of sensitivity to shifts.** We illustrate how downsampling affects shift-equivariance with a toy example. **(Top-Left)** An input toy signal is in light gray; max-pooled ($k = 2$, $s = 2$) toy signal is in blue. **(Top-Right)** Simply shifting the input and then max-pooling provides a completely different answer (red). **(Bot-Left)** The blue and red points are inherently sampled from densely max-pooled ($k = 2$, $s = 1$) intermediate signal (**thick black**). **(Bot-Right)** We instead sample from the *low-passed* intermediate signal, shown in green and magenta, better preserving shift-equivariance.

A representation is shift-invariant if shifting the input results in an *identical* representation.

$$\widetilde{\mathcal{F}}(X) = \widetilde{\mathcal{F}}(\text{Shift}_{\Delta h, \Delta w}(X)) \quad \forall\, (\Delta h, \Delta w) \tag{2}$$

For modern classifiers, layer $l = 0$ is the raw pixels, and final layer $L$ is a probability distribution over $D$ classes, $\mathcal{F}_L \in \Delta^{1 \times 1 \times D}$. The net typically progressively reduces spatial resolution, until all resolution is lost and features are of shape $\mathbb{R}^{1 \times 1 \times C_l}$. A common technique, such as used in (Lin et al., 2013; He et al., 2016; Huang et al., 2017), is to average across the entire convolutional feature map spatially, and use fully-connected layers in all subsequent layers, which can be expressed as $1 \times 1$ convolutions (Long et al., 2015). In such a setting, as proven by Azulay & Weiss (2018), shift-invariance on the output will necessarily emerge from shift-equivariance in the convolutional features.

**Modulo-N shift-equivariance/invariance** In some cases, the definitions in Equations 1, 2 may hold only when shifts $(\Delta h, \Delta w)$ are integer multiples of N. We refer to these scenarios as modulo-N shift-equivariance or invariance. For example, modulo-2 shift-invariance means that even-pixel shifts of the input result in an identical representation, but odd-pixel shifts may not.

### 3.2 Conventional Pooling vs Proposed Pool-Blur-Downsample

**Conventional strided pooling breaks shift-equivariance** In Fig. 3, we show an example 1-D signal $[0, 0, 1, 1, 0, 0, 1, 1]$. Max-pooling (kernel $k = 2$, stride $s = 2$) will result in $[0, 1, 0, 1]$. Simply shifting the input by one index results a dramatically different answer of $[1, 1, 1, 1]$. Shift-equivariance is lost. As seen in the bottom-left, both of these results are inherently downsampling from an intermediate signal – the input signal densely max-pooled ($k = 2$, $s = 1$). We can write a max-pooling layer as a composition of two functions, max-pooling densely evaluated, followed by naive downsampling: $\text{MaxPool}_{k,s}(X) = \text{Downsample}_s(\text{MaxPool}_{k,1}(X))$. Max-pooling preserves shift-equivariance (when evaluated densely), but naive downsampling does not.

**Blurring before downsampling better preserves shift-equivariance** We propose to low-pass filter the intermediate signal before downsampling, as shown in Fig. 3(bot-right). We define our MaxPool-BlurDownsample operator below.

$$\text{MaxPoolBlurDS}_{k,s}(X) = \text{Downsample}_s(\text{Blur}_{k_{blur}}(\text{MaxPool}_{k,1}(X))) \tag{3}$$

Sampling from the low-pass filtered signal gives $[.5, 1, .5, 1]$ and $[.75, .75, .75, .75]$ (Fig. 3 bot-right). These are closer to each other and better representations of the intermediate signal.

The method allows for a choice of blur kernel. In image processing, small kernels are often used across applications such as edge detection (Canny, 1986) and image pyramids (Adelson et al., 1984). We try a number of kernels, ranging from size $2 \times 2$ to $7 \times 7$. As the blur kernels are separable, it

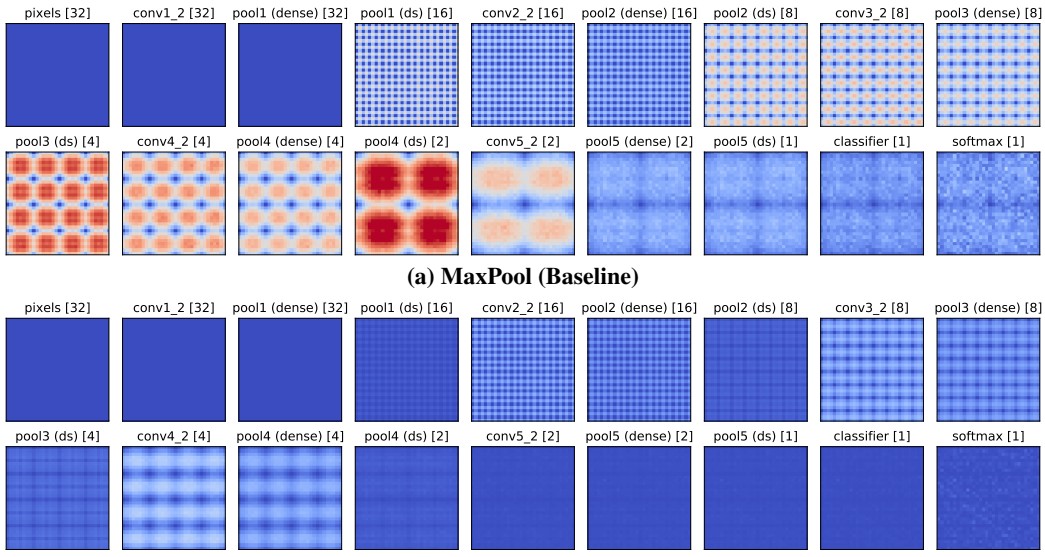

**(a) MaxPool (Baseline)**

**(b) MaxPoolBlurDownsample (Proposed)** with $5 \times 5$ **Triangle Low-Pass Filter**

**Figure 4: Shift-equivariance throughout the network.** We compute feature distance between left and right-hand sides of the shift-equivariance condition in Equation 1. Each point in each heatmap is a shift $(\Delta h, \Delta w)$. Layer resolution is in [brackets]; in the last three, shift-equivariance is equivalent to shift-invariance. Layers `pix-pool1(dense)` have perfect equivariance (distance 0 at all shifts, shown by blue). Red is half mean distance between two random *different* images, and is adjusted depending on the layer. **(a)** On the baseline, shift-equivariance is reduced each time downsampling takes place. Modulo-N shift-equivariance holds, with N doubling with each downsampling. **(b)** With our proposed change, shift-equivariance is better maintained, and the resulting classfication (softmax) layer is more shift-invariant.

can be implemented as a series of two convolutions (vertical blur followed by horizontal), and added computation scales linearly with $k_{blur}$, rather than quadratically.

## 4 EXPERIMENTS

### 4.1 EXPERIMENTAL SETUP

**Data, architecture, training schedule** We test on CIFAR10 classification (Krizhevsky & Hinton, 2009), which consists of 50k training and 10k testing images at resolution $32 \times 32$. We use the VGG13 architecture (Simonyan & Zisserman, 2014) from the PyTorch framework (Paszke et al., 2017)[1] and will make code available.

Each block consists of 2 `Conv-BatchNorm-ReLU` chunks, followed by `MaxPool`, doubling feature channels and halving spatial resolution until all resolution is lost. A final softmax predicts a probability vector. We use stochastic gradient descent (SGD) with momentum 0.9 and batch size 128. We train for 100 epochs at initial learning rate 0.1 and 50 additional epochs at 0.01 and 0.01.

**Low-pass filter kernels** We try a number of standard low-pass filters, shown in Table 1, ranging from size $2 \times 2$ to $7 \times 7$. All filters allow the DC signal pass and suppress (or completely kill) the highest frequency. Variations in filters correspond to tradeoffs between location of the cutoff frequency, slope of the cutoff, and variation of lobes in the passband and stopband. These properties are well-studied in the context of finite impulse response (FIR) filter design. However, it is unclear which types of filters are best suited for deep networks, so we empirically investigate their effects.

**Circular convolution and shifting** Edge artifacts are an important consideration. When an image is shifted, information is necessarily lost on one side, and has to be filled in on the other. In all our experiments, we use circular shifting and convolution. When the convolutional kernel hits the edge, it wraps to the other side. When shifting, pixels are "rolled" off the edge to the other side.

$$[\text{Shift}_{\Delta h, \Delta w}(X)]_{h,w,c} = X_{(h-\Delta h)\%H,(w-\Delta w)\%W,c} \text{ , where \% is the modulus function} \quad (4)$$

---

[1]https://github.com/pytorch/vision/blob/master/torchvision/models/vgg.py

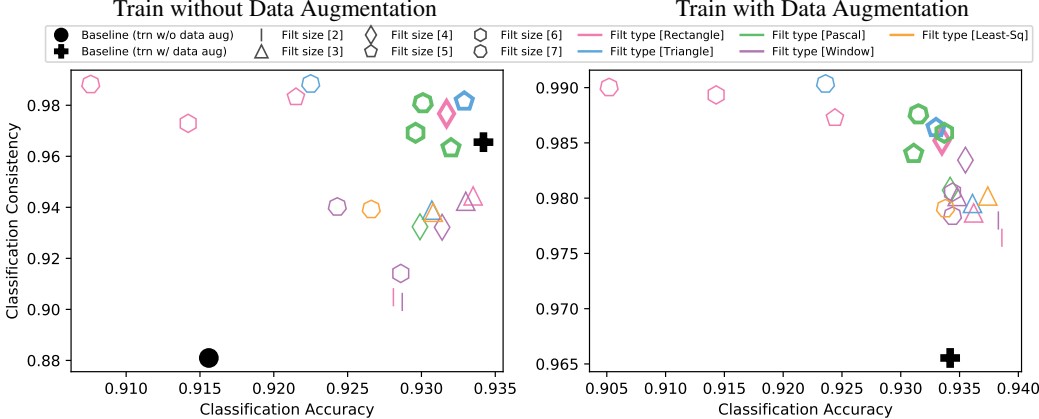

**Figure 5: Classification consistency vs. classification.** Networks trained **(left)** without and **(right)** with shift-based data augmentation, using various filters. Up (more consistent) and to the right (more accurate) is better. Number of sides corresponds to number of filter taps used (e.g., diamond for 4-tap filter); colors correspond to different methods for generating FIR filters. We highlight filters **Rectangle (4)**, **Triangle (5)**, and **Binomial (5-7)**, which perform consistently well in both metrics and settings.

| Filter shape | # Taps | Weights | Train with no augmentation | | | Train with augmentation | | |
|---|---|---|---|---|---|---|---|---|
| | | | Test accuracy | | Classification | Test accuracy | | Classification |
| | | | None | Rand | Consistency | None | Rand | Consistency |
| **Delta (baseline)** | 1 | [1] | 91.6 | 87.4 | 88.1 | 93.4 | 93.7 | 96.6 |
| **Rectangle** | 2 | [1, 1] | 92.8 | 89.3 | 90.5 | 93.9 | 93.8 | 97.6 |
| **Rectangle** | 3 | [1, 1, 1] | 93.4 | 91.8 | 94.5 | 93.6 | 93.7 | 97.9 |
| **Rectangle** | 4 | [1, 1, 1, 1] | 93.2 | 92.9 | 97.7 | 93.4 | 93.4 | 98.5 |
| **Rectangle** | 5 | [1, 1, 1, 1, 1] | 92.2 | 92.1 | 98.3 | 92.4 | 92.5 | 98.7 |
| **Rectangle** | 6 | [1, 1, 1, 1, 1, 1] | 91.4 | 91.2 | 97.3 | 91.4 | 91.5 | 98.9 |
| **Rectangle** | 7 | [1, 1, 1, 1, 1, 1, 1] | 90.8 | 90.7 | 98.8 | 90.5 | 90.5 | 99.0 |
| **Triangle** | 3 | [1, 2, 1] | 93.1 | 91.4 | 93.9 | 93.6 | 93.5 | 98.0 |
| **Triangle** | 5 | [1, 2, 3, 2, 1] | 93.3 | 93.0 | 98.2 | 93.3 | 93.2 | 98.6 |
| **Triangle** | 7 | [1, 2, 3, 4, 3, 2, 1] | 92.3 | 92.3 | 98.8 | 92.4 | 92.3 | 99.0 |
| **Binomial** | 4 | [1, 3, 3, 1] | 93.0 | 91.1 | 93.2 | 93.4 | 93.3 | 98.1 |
| **Binomial** | 5 | [1, 4, 6, 4, 1] | 93.2 | 92.6 | 96.3 | 93.1 | 93.2 | 98.4 |
| **Binomial** | 6 | [1, 5, 10, 10, 5, 1] | 93.0 | 92.4 | 96.9 | 93.4 | 93.2 | 98.6 |
| **Binomial** | 7 | [1, 6, 15, 20, 15, 6, 1] | 93.0 | 93.0 | 98.1 | 93.2 | 93.2 | 98.8 |
| **Window** | 3 | [1, 1.57, 1] | 93.3 | 91.5 | 94.2 | 93.5 | 93.5 | 98.0 |
| **Window** | 6 | [-1, 1.67, 5, 5, 1.67, -1] | 92.9 | 90.2 | 91.0 | 93.4 | 93.5 | 98.1 |
| **Window** | 7 | [-1, 0, 3, 4.71, 3, 0, -1] | 92.4 | 91.1 | 94.0 | 93.4 | 93.5 | 97.8 |
| **Least Squares** | 3 | [1, 1,63, 1] | 93.1 | 91.4 | 93.8 | 93.7 | 93.8 | 98.0 |
| **Least Squares** | 7 | [-1, 0, 3.80, 6.13, 3.80, 0, -1] | 92.7 | 91.0 | 93.9 | 93.4 | 93.5 | 97.9 |

**Table 1: Classification consistency and classification.** Results across blurring filters and training scenarios (without and with data augmentation). We evaluate classification accuracy without shifts (**Test accuracy – None**) and on random shifts (**Test accuracy – Random**), as well as **classification consistency**. Highlighted filters perform consistently well in both metrics and settings, as more easily seen in Fig. 5.

This modification minorly decreases classification performance, $93.8\%$ vs $93.4\%$ with data augmentation. This could potentially be mitigated by additional padding, at the expense of memory and computation. But more importantly, this methodology affords us a clean testbed. Any loss in shift-equivariance or invariance is purely due to characteristics of the feature extractor.

## 4.2 ANALYSIS

We measure shift-equivariance/invariance in three ways, targeting different aspects. We first focus on the shift-equivariance of the internal layers. We then check on the agreement of the hard output classification. Finally, we measure how much the soft predicted probability itself varies.

1. **Feature distance (lower is better).** We test how close shift-equivariance and invariance are to being fulfilled by computing $d(\text{Shift}_{\Delta h, \Delta w}(\widetilde{\mathcal{F}}(X)), \widetilde{\mathcal{F}}(\text{Shift}_{\Delta h, \Delta w}(X)))$ and $d(\widetilde{\mathcal{F}}(X), \widetilde{\mathcal{F}}(\text{Shift}_{\Delta h, \Delta w}(X))$ (left & right-hand sides of Eq. 1, 2), respectively. We use cosine distance, which is commonly used for deep features (Kiros et al., 2015; Zhang et al., 2018).

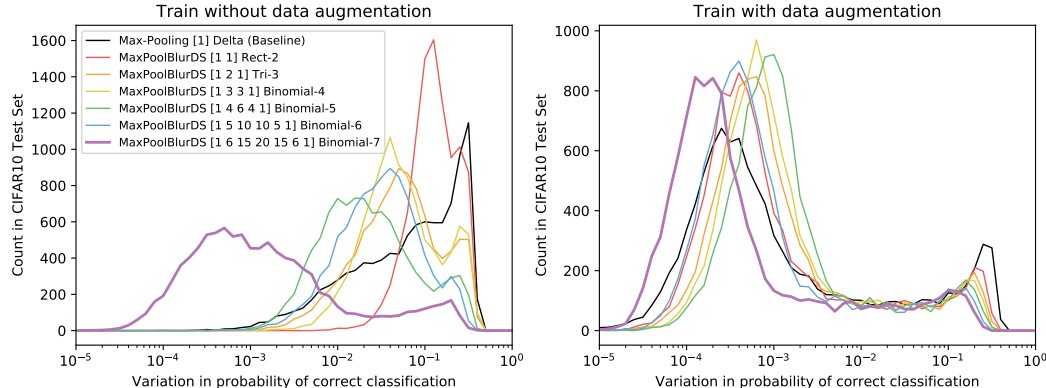

**Figure 6: Distribution of per-image classification variation.** We show the distribution of classification variation in the test set, **(left)** without and **(right)** with data augmentation at training. Lower variation means more consistent classifications (and increased shift-invariance). Training with data augmentation drastically reduces variation in classification. Adding filtering further decreases variation.

2. **Classification consistency (higher is better).** Perhaps of greatest interest is the actual decisions the classifier makes. We can measure its consistency by checking how often the network outputs the same classification, given the same image with two different shifts: $\mathbb{E}_{(X,h_1,w_1,h_2,w_2)}\mathbb{1}\{\arg\max P(\text{Shift}_{h_1,w_1}(X)) = \arg\max P(\text{Shift}_{h_2,w_2}(X))\}$.

3. **Classification variation (lower is better).** The metric above looks at the hard classification, discounting classifier confidence. Similar to Azulay & Weiss (2018), we trace the variation in probability of correct classification, given different shifts. We can capture the variation across all possible shifts: $\sqrt{Var_{h,w}(\{P_{\text{correct class}}(\text{Shift}_{h,w}(X))\}\})}$.

Table 1 shows results across a number of different low-pass filters, training with and without data augmentation. We dissect the results below.

**How shift-equivariant are deep features?** In Fig. 4 (top), we compute distance from shift-equivariance, as a function of all possible shift-offsets $(\Delta h, \Delta w)$ and layers. `MaxPool` layers are broken into two components – before and after downsampling. Pixels are trivially shift-equivariant, as are all layers before the first downsampling. Once downsampling occurs in `pool1(ds)`, shift-equivariance is lost. However, modulo-N shift-equivariance still holds, and each subsequent downsampling doubles the factor.

Additionally, we observe that before the downsampling operation, the pooling layer first increases shift-equivariance (e.g., `conv3_2` to `pool3(dense)`). This is consistent with the long-held intuition that pooling build invariances inside the network (LeCun et al., 1990) and isolates the downsampling operation as the culprit behind loss of shift-equivariance.

**Does blurring before downsampling achieve better shift-equivariance?** In Fig. 4 (bottom), we add a blurring filter to the `MaxPool` layers, as proposed in Section 3, and again plot shift-equivariance maps for each layer. Shift-equivariance is clearly better preserved. In particular, the severe drop-offs in downsampling layers do not occur. Improved shift-equivariance throughout the network cascades into more consistent classifications in the final softmax layer.

Some selected examples are in Fig. 2. Our method stabilizes the classifications. In Fig. 6, we show the distribution of classification variations, before and after adding in the low-pass filter. Even a small $2 \times 2$ filter, immediately variation. As the filter size is increased, the output classification variation decreases. This has a larger effect when training without data augmentation, but is still observable when training with data augmentation.

**Does shift-invariance degrade performance?** Our method produces more shift-equivariant feature maps and consequently, more shift-invariant outputs. However, does this come at a cost?

We study the output classification consistency versus classification accuracy. In Fig. 5 (left), we show results, trained without shift-based data augmentation. Training with the baseline MaxPooling gives accuracy 91.6% and consistency 88.1%. Our proposed change – with a $5 \times 5$ triangle filter improves accuracy to 93.3% and consistency to 98.2%. This indicates that low-pass filtering does

not destroy the signal, or make learning harder. On the contrary, preserving shift-equivariance serves as "built-in" augmentation, indicating more efficient data usage.

In principle, networks can *learn* to be shift-invariant from data. Does adding shift-based data augmentation remove the benefit from method? Shift-based data augmentation with the baseline network results in consistency of $96.6\%$, lower than our method trained *without* data augmentation. In addition, as seen in Fig. 5 (right), applying out method *with* data augmentation provides an immediate jump in classification consistency, while maintaining accuracy. From there, a clear tradeoff appears – higher amounts of shift-invariance can be achieved at the cost of decreased accuracy. For example, very large rectangular filters over-aggressively smooth the signal. Downstream applications may favor one factor over another, and the choice of filter allows one to explore this space.

Fig. 6 investigates the distribution of classification variations. Training with data augmentation with the baseline network reduces variation (black lines on both plots). Our method reduces variation in both scenarios. More aggressive filtering further decreases variation.

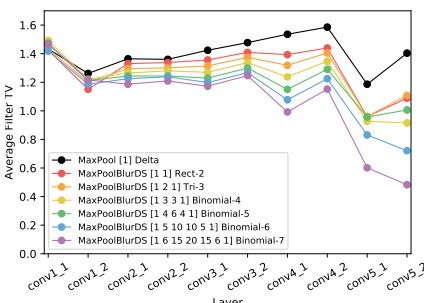

**Figure 7: Total Variation (TV) by layer.** We compute average smoothness of learned conv filters per layer (lower is smoother). Baseline MaxPool is in black, and adding additional blurring is shown in colors. Note that the *learned* convolutional layers become smoother, indicating that a smoother feature extractor is induced. The **Binomial-7** filter produces consistently strong results, in both consistency and accuracy.

**How do the learned convolutional filters change with the proposed modification?** We measure spatial smoothness using the normalized Total Variation (TV) metric proposed in Ruderman et al. (2018). Our proposed change smooths the internal feature maps for purposes of downsampling. As shown in Fig. 7, this induces smoother *learned* filters throughout the network. Adding in more aggressive blur kernels further decreases the TV (increasing smoothness). This indicates that our modification actually induces a smoother feature extractor overall.

**How does the proposed method affect timing?** In Tab. 2, we show the added time each element of the proposed method takes: evaluating the MaxPool layer at stride 1 instead of stride 2, and running a blurring filter. Since the blurring filters are separable, time increases linearly with filter size. The largest filter adds $12.3\%$ per forward pass. This is significantly cheaper than evaluating multiple forward passes in an ensembling approach ($1024\times$ computation to evaluate every shift), or evaluating each layer more densely by exchanging striding for

| Model | Timing [ms] | % added |
|---|---|---|
| Baseline | 10.19 | +0.00% |
| + dense pool | 10.50 | +3.04% |
| + dense pool + $3 \times 3$ filter | 11.06 | +8.52% |
| + dense pool + $5 \times 5$ filter | 11.27 | +10.6% |
| + dense pool + $7 \times 7$ filter | 11.45 | +12.3% |

**Table 2: Timing analysis** We test the average speed of a forward pass on a GTX1080Ti Nvidia GPU for a batch size of 100 of $32 \times 32$ image with the VGG13 network.

dilation ($4\times, 16\times, 64\times, 256\times$ computation for `conv2-conv5`, respectively). These timings are on our VGG13 network setup. With deeper networks, the relative added computation decreases.

## 5 CONCLUSIONS AND DISCUSSION

In summary, we show that shift-invariance is lost through a deep network, as downsampling in pooling layers do not meet the Nyquist criteria. We propose a simple architectural modification, following signal processing principles, to improve shift-equivariance. This change allows the network architecture designer to keep their pooling layer of choice untouched.

We achieve higher consistency while maintaining classification performance. In addition, we show large improvements in both performance and consistency when training without data augmentation. This is potentially applicable to online learning scenarios, where the data distribution is changing. Future directions include exploring the potential benefit to downstream applications, such as nearest-neighbor retrieval, improving temporal consistency in video models, robustness to adversarial examples, and high-level vision tasks such as detection. Another possible future direction is learning the downsampling kernels. Overall, our experiments indicate that this classical signal processing technique has a place in modern deep networks.

APPENDIX

## A   DenseNet architecture

Blurring before downsampling can be applied to any strided layer in any network. We provide an additional experiment using the DenseNet architecture (Huang et al., 2017). In Fig. 8, we show classification consistency vs. accuracy, similar to Fig. 5 for VGG13 in the main paper.

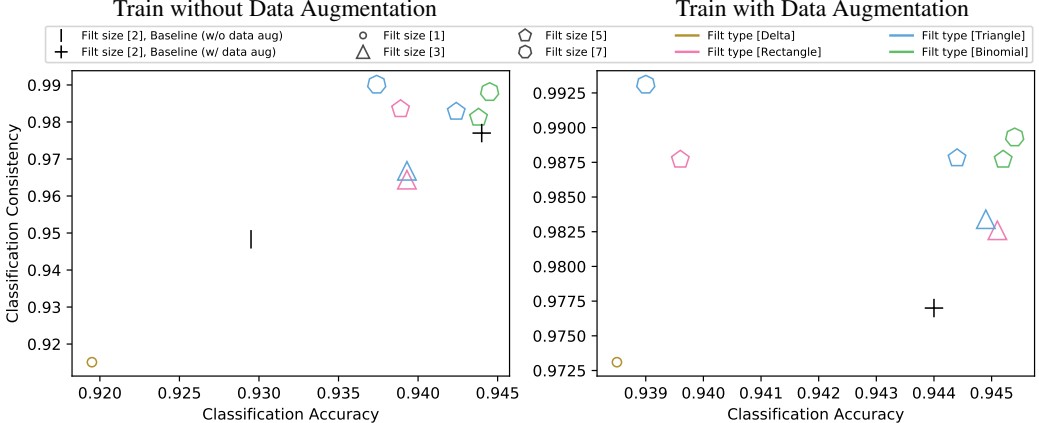

**Figure 8: Classification consistency vs. classification for DenseNet** Same test in as in Fig. 5, but with DenseNet (Huang et al., 2017) instead of VGG13 (Simonyan & Zisserman, 2014). We show networks trained **(left)** without and **(right)** with shift-based data augmentation, using various blurring filters. Consistency is computed by computing classification of an image with two random shifts, and checking for agreement. Up (more consistent) and to the right (more accurate) is better. Number of sides corresponds to number of filter taps used (e.g., triangle for 3-tap filter); colors correspond to different methods for generating FIR filters.

**Comparison to VGG13 for Baseline Network**   We use the DenseNet-40-12 architecture, from a publicly available implementation.[2] Relative to VGG13, DenseNet achieves higher performance ($94.4\%$ vs $93.8\%$ ), despite using fewer parameters (1M vs 9M). DenseNet also starts with higher shift-invariance ($97.7\%$ vs. $96.6\%$) for two reasons: (a) fewer downsampling layers (2 vs 5) and (b) already using blurring before downsampling, in the form of `AveragePool` layers, equivalent to using a **Rectangle (2)** filter. We investigate the effects of replacing this $2 \times 2$ filter.

**Results on DenseNet**   Our method improves the DenseNet results, and confirms the findings in the main paper. In some cases, results are actually stronger. The primary findings are:

- As seen in Fig. 8 (left), using a stronger low-pass filter, such as **Binomial (5, 7)** *without* data augmentation, provides competitive performance compared to the baseline trained *with* data augmentation. For **Binomial (7)**, performance is actually better in both consistency and accuracy.
- As seen in Fig. 8 (right), when training with data augmentation, using filters such as **Rect (3)**, **Triangle (3,5)**, and **Binomial (5,7)** not only increase consistency, as expected, but also slightly increases accuracy, surprisingly.

The observations for DenseNet corroborate the results from VGG13 in the main paper, further demonstrating the effectiveness of blurring before downsampling.

## B   Robustness to shift-based adversary

In the main paper, we show that using the proposed **PoolBlurDownsample** method increases the classification consistency, while maintaining accuracy. A logical consequence is increased accuracy in presence of a shift-based adversary. We empirically confirm this in Fig. 9 for VGG13 on CI-FAR10. We compute classification accuracy as a function of maximum adversarial shift. A max

---

[2]https://github.com/andreasveit/densenet-pytorch

shift of 2 means the adversary can choose any of the 25 positions within a $5 \times 5$ window. For the classifier to "win", it must correctly classify all of them correctly. Max shift of 0 means that there is no adversary. Conversely, a max shift of 16 means the image must be correctly classified at all $32 \times 32 = 1024$ positions.

Our primary observations are as follows:

- As seen in Fig. 9 (left), the baseline network (gray) is very sensitive to the adversary.
- Adding larger **Binomial** filters (from red to purple) increases robustness to the adversary. In fact, **Binomial (7)** filter (purple) *without* augmentation outperforms the baseline (black) *with* augmentation.
- As seen in Fig. 9 (right), adding larger **Binomial** filters also increases adversarial robustness, even when training with augmentation.

These results corroborate the findings in the main paper, and demonstrate a use case: increased robustness to shift-based adversarial attack.

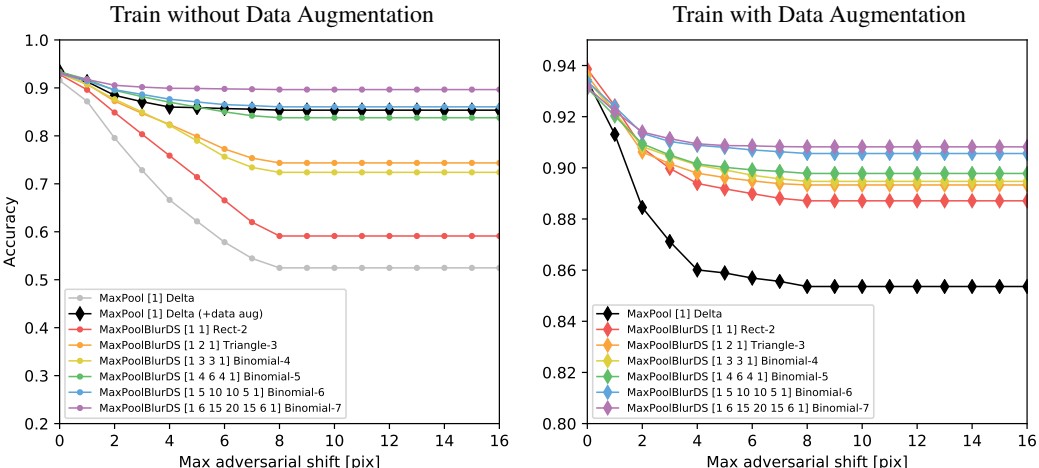

**Figure 9: Robustness to shift-based adversarial attack.** Classification accuracy as a function of the number of pixels an adversary is allowed to shift the image. Applying our proposed filtering increases robustness, both without **(left)** and with **right** data augmentation.

## C  EFFECT OF BLURRING BEFORE POOLING

In our proposed method, we break the strided-pooling operation into two, and blur in between. This allows us to directly blur before downsampling, which has solid theoretical backing in sampling theory (Oppenheim et al., 1999), and is commonly used in image processing (Gonzalez & Woods, 1992), graphics (Foley et al., 1995), and computer vision (Szeliski, 2010). Here, we empirically investigate blurring before pooling instead.

Fig. 10 shows the results by applying blurring first (shown in the gray points), in comparison our proposed method (colored polygons, as shown before in Fig. 5). We make the following observations:

- In Fig. 10 (right), when training with augmentation, *blurring before pooling reduces performance for all filters*. For almost all filters (with few exceptions), both classification accuracy and consistency are significantly reduced.
- In Fig. 10 (left), when training without augmentation, the lower performing filters actually perform better when blurring before filtering. For the better filters, however, *blurring before pooling lowers performance* (similar shift-invariance, but lower accuracy).

The signal pre-pooling is undoubtedly related to the signal post-pooling. Thus, blurring before pooling provides "second-hand" anti-aliasing, and still increases shift-invariance over the baseline. Though it does empirically help in certain circumstances (the lower-performing filters, without augmentation), the best performing filters use the proposed **PoolBlurDownsample** ordering.

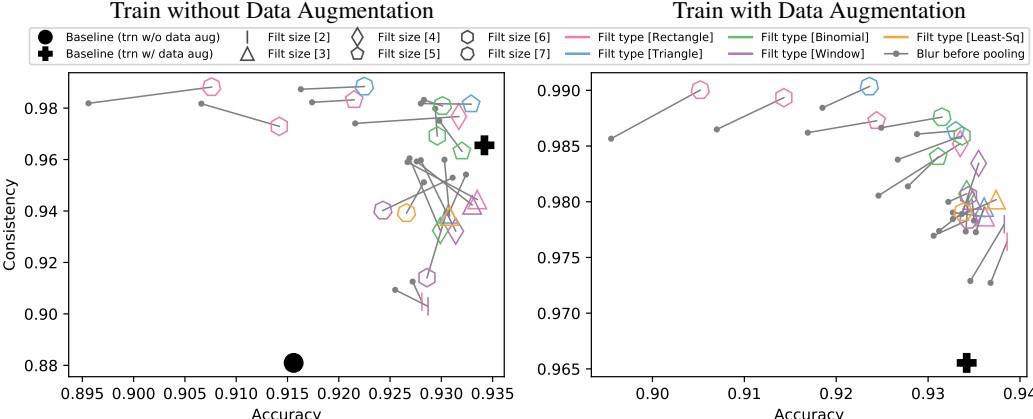

**Figure 10: Blurring before pooling.** Blurring before the max-pooling (gray points) for different filters, as compared to their PoolBlurDownsample counterparts (colored polygons). The poorer performing filters, when training without data augmentation, observe an increase in performance. For almost all filters when training with data augmentation, and for the higher-performing filters training without data augmentation, performance is significantly reduced, often in both accuracy and consistency. Directly blurring the downsampled signal (after the pooling layer), as proposed in the main paper, is more effective.

# D  AVERAGE ACCURACY ACROSS SPATIAL POSITIONS

In Figure 11, we show how accuracy systematically degrades as a function of spatial shift, when training without augmentation. We observe the following:

- On the left, the baseline heatmap shows that classification accuracy when testing with no shift, but quickly degrades when shifting.
- The proposed filtering decreases the degradation. **Binomial-7** is largely consistent across all spatial positions.
- On the right, we plot the accuracy when making diagonal shifts. As increased filtering is added, classification accuracy becomes consistent in all positions.

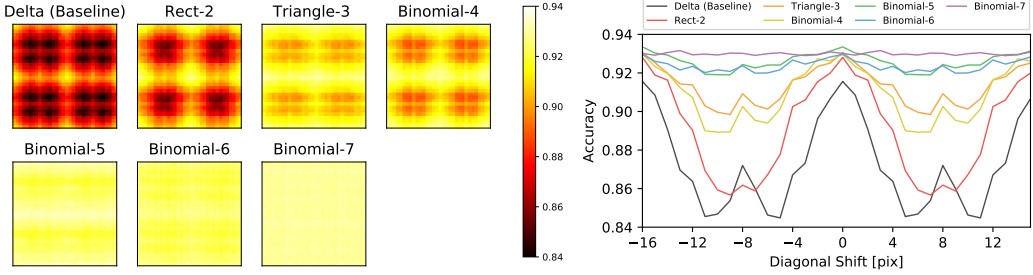

**Figure 11: Average accuracy as a function of shift. (Left)** We show classification accuracy across the test set as a function of shift, given different filters. **(Right)** We plot accuracy vs diagonal shift in the input image, across different filters. Note that accuracy degrades quickly with the baseline, but as increased filtering is added, classifications become consistent across spatial positions.

# E  FILTER DISCUSSION

## E.1  FILTER SELECTION

We use select standard low-pass filters to empirically test. The filter weights are shown in Tab. 1 in the main paper. Note that weights are normalized to sum to 1. **Rectangle**, **Triangle**, and **Binomial** filters are discussed in textbooks such as (Szeliski, 2010). **Window** and **Least Squares** are more advanced FIR filter design techniques.

- **Rectangle**: a moving average, often referred to as a box filter. The filter is a vector of length ones. For example, **Rect-2**, is $[1, 1]$. This filter, followed by subsampling, is equivalent to the `AveragePooling` layer.

- **Triangle**: linearly decreases weight of neighboring values. This is equivalent of applying box filtering twice. For example, **Triangle-3** is $[1, 2, 1]$, is two **Rect-2** $[1, 1]$ filters convolved together, and **Triangle-5** is $[1, 2, 3, 2, 1]$, is two **Rect-3** $[1, 1, 1]$ filters convolved together.

- **Binomial**: Filter used in Laplacian Pyramids (Burt & Adelson, 1987); $[1, 1]$ filter convolved with itself repeatedly. Note that **Binomial-2,3** is equivalent to **Rectangle-2** and **Triangle-3**, respectively.

- **Window**: filter produced using the window method (`firwin`), as described in "7.4 Optimum Approximations for FIR Filters" in Oppenheim et al. (1999).

- **Least Squares**: least squares error minimization (`firls`), from Python `scipy.signal` toolbox, as described in Selesnick (2005).

### E.2   FILTER SEPARABILITY

As discussed in Section 3.2, our filters are separable. Consider a $G \in \mathbb{R}$. If $G$ is rank-1, it can be decomposed (or separated) into $G_y G_x$, where $G_y \in \mathbb{R}^{K \times 1}$ and $G_x \in \mathbb{R}^{1 \times K}$. This is an important consideration when convolving $G$ with signal $X \in \mathbb{R}^{H \times W}$.

$$G * X = (G_y * G_x) * X = G_y * (G_x * X), \text{where} * \text{is convolution} \tag{5}$$

The left-hand side, evaluating the blur with 2-D convolution takes $H \times W \times K \times K$ multiply-adds, with runtime scaling quadratically with $K^2$. Meanwhile, evaluating a horizontal and vertical blur sequentially takes $H \times W \times K$ multiply-adds each, scaling *linearly* by $K$.

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
