# OpenReview forum: "Making Convolutional Networks Shift-Invariant Again"
_ICLR.cc/2019/Conference_

### Official Review · AnonReviewer2 · 2018-11-02
**Interesting approach in its simplicity with flaws in evaluation**

**Rating:** 5
**Confidence:** 4

**Review:**

This work shows that adding a simple blurring into max pooling layers can address issues of image classification instability under small image shifts. In general this work presents a simple and easy to implement solution to a common problem of CNNs and even though it lacks more thorough theoretical analysis of this problem from the signal processing perspective (such as minimal size of the blurring kernel for fulfilling the Nyquist-Shannon sampling theorem), it seems to provide ample empirical evidence.

Pros:
+ The introduction and motivation is really well written and Figure 3 provides a clear visualisation main max pooling operator issues.
+ The proposed method is really simple and shows promising results on the CIFAR dataset. With random shifts, authors had to tackle cropping with circular shifts. As it can cause artifacts in the data, authors also provide baseline performances on the original data (used for both training and testing).
+ Authors provide a thorough evaluation, ranging from comparing hidden representations to defining consistency metrics of the classified classes.

This work is lacking in the experimental section due to some missing details and few inconsistencies. I believe the most of my concerns can be relatively easily fixed/clarified in an update of this submission.

Major issues, which if fixed would improve the rating:
- It is not correct to average test accuracy and test consistency as both measures are different quantities, especially when using them for ranking. The difference between accuracy of different methods are considerably smaller than differences in the classification consistency.
- It is not clear how many shifts are used for computing the "Random Test Accuracy" and the "Classification Accuracy". Also whether the random shifts are kept constant between evaluated networks and evaluation metrics.
- Authors do not address the question what is the correct order of operations for the blurring. E.g. would the method empirically work if blurring was applied before max pooling? Do the operations commute?
- The selection of the filters is rather arbitrary, especially regarding the 1D FIR filters. The separability of these filters should be discussed.
- I believe authors should address how this work differs to [1], as it also tests different windowing functions for pooling operators, even though in different tasks.

Minor issues, which would be nice to fix however which do not influence my rating:
* Section 3.1 - And L-Layer deep *CNN*, H_l x W x C_l -> H_l x W_l x C_l
* Section 3.1. Last paragraph - I would not agree with the statement that in CNNs the shift invariance must necessarily emerge upon shift equivariance. If anything, this may hold only for the last layer of a network without fully connected layers and with average pooling of the classifier output (ResNet/GoogleNet like networks).
* Explicitly provide the network architecture as [Simonyan14] does not test on CIFAR and cannot use Batch normalisation.
* It would be useful to add citation for the selected FIR filters.
* The flow of section 4.2. can be improved to help readability. The three metrics should be first motivated before their introduction. Metric 2. paragraph - the metric is defined below, not above.
* It would be interesting to see what would be the performance if the blurring filters were trained as well (given some sensible initialisation).
* One future direction would be to verify that this approach generalises to larger networks as well. It might be worth to discuss this in the conclusions.

[1] Scherer, Dominik, Andreas Müller, and Sven Behnke. "Evaluation of pooling operations in convolutional architectures for object recognition." Artificial Neural Networks–ICANN 2010. Springer, Berlin, Heidelberg, 2010. 92-101.

---

> ### Author Response · Authors · 2018-11-27
> **(Part I) Added experiment confirms correct pool/blur ordering; DenseNet experiment corroborates results; clarifications added**
>
> We thank the reviewer for the detailed comments. We are happy that the reviewer found the motivation “really well written”, the method simple, and the results promising. We have updated the draft. We address all major and minor comments from the review below. In particular, we provide new experiments on switching blurring & pooling, as well as on DenseNet. These results back up the findings in the original submission.
>
> ----- ADDITIONAL EXPERIMENTS -----
> > “Authors do not address the question what is the correct order of operations for the blurring. E.g. would the method empirically work if blurring was applied before max pooling? Do the operations commute?”
>
> Thank you for this suggestion. We have added this experiment in Appendix C and Figure 10. Note that our proposed method applies the signal to the exact signal which is to be downsampled, which has solid theoretical backing in sampling theory (Oppenheim et al., 1999). The operations to not commute (as max-pooling is nonlinear). Switching the ordering separates the blurring from downsampling, only providing “second-hand” blurring.
>
> Interesting, some of the worse-performing filters do improve when training without data augmentation. However, for the better-performing filters, performance is reduced. All filters perform worse when training with data augmentation.
>
> These experiments empirically confirm that the proposed PoolBlurDownsample method was the correct order of operations.
>
> > “(Minor) One future direction would be to verify that this approach generalises to larger networks as well. It might be worth to discuss this in the conclusions.”
>
> Thank you for the suggestion. In Appendix A and Figure 8, we show the results applied to a more modern DenseNet (Huang et al., 2017) architecture. The results confirm the findings from the VGG architecture. In short:
> - When training without data augmentation, the proposed technique improves both classification consistency and accuracy over the baseline (as before).
> - The proposed technique trained without data augmentation outperforms the baseline, even with data augmentation.
> - When training with data augmentation, the proposed technique improves consistency, and surprisingly, even slightly improves accuracy in this setting.
>
> These results support the general applicability of the method to CNNs.
>
> > “(Minor) It would be interesting to see what would be the performance if the blurring filters were trained as well.”
> We noted this direction in the discussion section of the submission and are currently investigating this interesting direction.
>
> ----- CLARIFICATIONS -----
> > “It is not clear how many shifts are used for computing the "Random Test Accuracy" and the "Classification Accuracy". Also whether the random shifts are kept constant between evaluated networks and evaluation metrics.”
>
> We have updated “Random Test Accuracy” to use every shift (all 32x32=1024 positions) for all 10k test images. “Classification Consistency” we test classification agreement between 10 randomly shifted pairs for each test image. This provides 100k examples total (standard error of ~0.05%-0.1%).
>
> > "It is not correct to average test accuracy and test consistency as both measures are different quantities, especially when using them for ranking."
>
> Thank you for pointing this out. We agree and have removed the averaging. The two factors -- classification and shift-invariance -- should be evaluated as separate dimensions.
>
> > “The selection of the filters is rather arbitrary, especially regarding the 1D FIR filters.”
> > Minor: “It would be useful to add citation for the selected FIR filters.”
>
> In Appendix E, we clarify the selected filters, which are from common references (textbooks) and toolboxes (scipy.signal Python) and add citations. “Rectangle” is a simple box filter. “Triangle”, and “Binomial” (which we renamed from Pascal) can be seen in Table 3.4 from (Szeliski, 2010) textbook [2]. “Binomial” was used in Image Pyramids [3]. “Window” [4] and “Least Squares” [5] are more modern filter design methods, implemented in FIR filter design toolboxes.
>
> > “The separability of these filters should be discussed.”
>
> We add a discussion of separability to Appendix E. In particular, we show that separability allows added computation to scale linearly, rather than quadratically, with filter size.
>
> ----- REFERENCES -----
> [1] Scherer, Dominik, Andreas Müller, and Sven Behnke. "Evaluation of pooling operations in convolutional architectures for object recognition." ICANN 2010.
> [2] Szeliski. Computer Vision: Algorithms and Applications. 2010.
> [3] Burt and Adelson, Laplacian Pyramid as a compact image code. IEEE Transactions on Communications. 1983.
> [4] Oppenheim and Schafer, "Discrete-Time Signal Processing". 2nd ed. 1999.
> [5] Selesnick. Linear-Phase Fir Filter Design By Least Squares. OpenStax CNX. Aug 9, 2005. http://cnx.org/contents/eb1ecb35-03a9-4610-ba87-41cd771c95f2@7

---

> ### Author Response · Authors · 2018-11-27
> **(Part II) Writing**
>
> ----- WRITING -----
> We address all writing suggestions from the review below.
>
> > “I believe authors should address how this work differs to [1], as it also tests different windowing functions for pooling operators, even though in different tasks.”
>
> Thank you for the reference. We add this reference in the final paragraph in related work. The work from [1] provides a systematic evaluation of different blurred-downsampling, similar to our work. However, it operates from the assumption that max-pooling and blurred-downsampling are strictly alternatives, and makes the recommendation to use max-pooling. We show that they are actually compatible! Advantages to max-pooling are kept while incorporating proper blurred-downsampling.
>
> > “(Minor) The flow of section 4.2. can be improved to help readability. The three metrics should be first motivated before their introduction. Metric 2. paragraph - the metric is defined below, not above.”
> Thank you for the suggestion. We have added a short motivational introduction in 4.2, and clarified the relation between metric 2 from metric 1.
>
> > “(Minor) Explicitly provide the network architecture as [Simonyan14] does not test on CIFAR and cannot use Batch normalisation.”
> We have added the reference in a footnote to the reference implementation, and will release code.
>
> > “(Minor) Section 3.1 - And L-Layer deep *CNN*, H_l x W x C_l -> H_l x W_l x C_l”
> Thank you for finding the typo.
>
> > “(Minor) Section 3.1. Last paragraph - I would not agree with the statement that in CNNs the shift invariance must necessarily emerge upon shift equivariance. If anything, this may hold only for the last layer of a network without fully connected layers and with average pooling of the classifier output (ResNet/GoogleNet like networks).”
>
> Our claim is that global average pooling of a shift-equivariant extractor results in a shift-invariant extractor. The proof is found in (Azulay and Weiss, 2018), starting from bottom of page 5. We sketch the proof below:
>
> Assume:
> (1) F(Shift(X)) = Shift(F(X)), feature extractor F is shift-equivariant
> (2) G = GlobAvgPool o F, global average pool after shift-equivariant feature extractor
>
> We wish to show that:
> (3) G(Shift(X)) = G(X), feature extractor G is shift-invariant
>
> Proof:
> G(Shift(X))
> = GlobAvgPool(F(Shift(X)), substitute (2) definition of G
> = GlobAvgPool(Shift(F(X)), substitute (1) shift-equivariance of F
> = GlobAvgPool(F(X)), shifting the feature map does not change its average
> = G(X), recombine using (2) definition of G
>
> From there, G(X) is a feature vector with no spatial extent. Any subsequent function with G serving as a front-end (even fully connected layers) will maintain shift-invariance, since G(Shift(X)) = G(X), wiping away the effect of any shift.
>
> Many networks, such as the VGG13 and DenseNet in our tests use global average pooling, followed by a single linear layer (and softmax) for classification.

---

### Official Review · AnonReviewer3 · 2018-11-02
**A paper with technical details and analysis, but the problem addressed does not seems to be interesting and significant**

**Rating:** 5
**Confidence:** 4

**Review:**

This paper analyzed on the core factor that make CNNs fail to hold shift-invariance, the naive downsampling in pooling. And based on that the paper proposed the modified pooling operation by introducing a low-pass filter which endows a shift-equivariance in the convolution features and consequently the shift-invariance of CNNs.

Pros:
1.	The paper proposed a simple but novel approach to make CNNs shift-invariant following the traditional signal processing principle.
2.	This work gave convincing analysis (from both theoretical illustrations and experimental visualizations) on the problem of original pooling and the effectiveness of the proposed blur kernels.
3.	The experiment gave some promising results. Without augmentation, the proposed method shows higher consistency to the random shifts.

Cons:
1.	When cooperating with augmentation, the test accuracy on random shifted images of proposed method did not exceed the baseline. Although the consistency is higher, it is secondary to the test accuracy of random shifted data. And it is confused to do average on consistency and test accuracy, which are in different scales, and then compare the overall performance on the averages.
2.	It seems to be more convincing if the ‘random’ test accuracy is acquired by averaging several random shifts on a single image and then do average among images, as well as to show how accuracy various on shifting distance.
3.	Some other spatial transforming/shifting adaptive approaches should be taken into consideration to compare the performance.
4.	There are some minor typos, such as line 3 in Section 3.1 and line 15 in Section 3.2

---

> ### Author Response · Authors · 2018-11-27
> **Simple proposal, derived from first principles, to fix a fundamental property is important**
>
> We thank the reviewer for the comments. We are happy the reviewer found the paper “simple but novel”, the analysis “convincing” and experiments “promising”. We first address the review title and clarify the goal of the paper. and then address individual points.
>
> > “problem addressed does not seems to be interesting and significant”
>
> Adversarial attack and defense is a large area of interest -- [1] has 1646 citations in 5 years, according to Google scholar. Lack of shift-invariance in modern deep networks exposes it to a very simple attack. We add an additional experiment, demonstrating practical use - robustness in presence of a shift-based adversarial attack.
>
> Blurring before downsampling is “textbook material” from sampling theory [2], image processing [3], computer graphics [4], and computer vision [5]. Proposing a fix from first-principles for a fundamental low-level problem (with implications on adversarial attacks/defenses) should be important. In the updated draft, we add these references to better clarify the fundamental nature of the proposed fix.
>
> > “test accuracy on random shifted images of proposed method did not exceed the baseline....consistency is secondary to the test accuracy...”
>
> In Appendix B, we show how consistency affects test accuracy. We compute classification accuracy, as a function of maximum adversarial shift. A max shift of 2 means the adversary can choose any of the 25 positions within a 5x5 window. For the classifier to “win”, it must classify all positions correctly. More detailed discussion is in Appendix B. In short:
> - The baseline is very sensitive to the adversary. Our proposed method dramatically decreases sensitivity to the adversary.
> - Again, our proposed method (with Binomial-7 filter) without augmentation is more robust than the baseline, trained with data augmentation.
>
> These results corroborate the findings in the main paper, and demonstrate a use case: increased robustness to shift-based adversarial attack.
>
> > “show how accuracy various on shifting distance.”
>
> Thank you for the suggestion. In Appendix D, we show how accuracy in the test set varies with shifted distance. The baseline accuracy drops quickly, but the proposed fix maintains classification accuracy across spatial shifts.
>
> > “other spatial transforming/shifting adaptive approaches should be taken into consideration to compare the performance.”
>
> Our paper focuses on thoroughly evaluating and incorporating shift-invariance. A feature extractor should first be robust to shifts in order to be robust to other spatial transforms, such as warps. In Appendix A, we further establish the effectiveness of our technique by testing on the DenseNet architecture.
>
> ----- CLARIFICATIONS -----
> > “And it is confused to do average on consistency and test accuracy, which are in different scales, and then compare the overall performance on the averages.”
>
> Thank you for pointing this out. We agree and have removed the averaging. The two factors -- classification and shift-invariance -- should be evaluated as separate dimensions.
>
> > “It seems to be more convincing if the ‘random’ test accuracy is acquired by averaging several random shifts on a single image and then do average among images”
>
> We provide wall-clock analysis in Table 2, showing that our fix adds +8-12% computation. Even evaluating twice would add +100% computation. Our goal is to better preserve shift-equivariance in the network, given roughly the same computation budget, and minimal perturbation to network architecture (as described in Section 2).
>
> ----- WRITING -----
> > “4. There are some minor typos, such as line 3 in Section 3.1 and line 15 in Section 3.2”
>
> Thank you for finding the typos. They are fixed in the updated draft.
>
> ----- REFERENCES -----
> [1] Szegedy et al. Intriguing properties of neural networks. ArXiv, 2013.
> [2] Section 4.6.1: Sampling Reduction by an Integer Factor. Oppenheim, Schafer, Buck. Discrete-Time Signal Processing. 2nd ed. 1999
> [3] Section 2.4.5: Zooming and Shrinking Digital Images. Gonzalez and Woods. Digital Image Processing. 2nd ed. 1992.
> [4] Section 14.10.6: Antialiasing in Practice. Foley, van Dam, Feiner, Hughes. Computer Graphics: Principles and Practice. 2nd ed. 1995.
> [5] Section 3.5.2: Decimation. Szeliski. Computer Vision: Algorithms and Applications. 2010.

---

### Official Review · AnonReviewer1 · 2018-11-03
**Making CNNs translation equivariant again, potentially important line of work with multiple loose ends**

**Rating:** 6
**Confidence:** 4

**Review:**


Summary

From a theoretical point of view, one might be tempted to believe that deep CNNs are translation equivariant and their predictions are translation invariant. In practice, this is not necessarily true. The authors propose to augment standard deep CNNs with low-pass filters to reduce this problem. The results seem promising for an older VGG architecture.

Quality

The paper is very verbose, the figures and captions are tedious to read, the mathematical notation seems strange as well, making the writing more concise is highly encouraged. The main ideas are easy to follow and the choice of experiments seems fine.

Significance

This is the first empirical work trying to fix the issue of non-translation equivariance in convolutional neural networks. The conclusions of this work are potentially relevant for a wide audience of CNN practitioners.

Main Concerns

To show that all claims of the paper do indeed hold, the authors should attack their augmented network with the translation attack of [1]. As robustness to this type of transformations is one of the main goals, it should be tested if it was achieved. The attack can be found in some open source frameworks [2] and should be easy to apply.

Wall-clock times need to be reported for the various blurring kernels and compared to the baselines.

Extend results to a cutting-edge architecture, e.g. DenseNets or Wide ResNets. If this result is not provided the significance of the work is not clear.

Despite being more expensive, do dilations fix the issue of missing translation equivariance provably and not just approximately like the low-pass filtering approach proposed here? This should be discussed and a comparison in terms of wall-clock time would be great as well.

Minor

- Strange notation e.g. in equation 1. Why not write: x+\delta x in the argument of the function instead of "Shift". The current notation seems unnecessarily informal.
- Figure 4: show scale and color bar.

[1] Engstrom et al., "A rotation and a translation suffice: Fooling cnns with simple transformations."
[2] https://foolbox.readthedocs.io/en/latest/modules/attacks/decision.html#foolbox.attacks.SpatialAttack

---

> ### Author Response · Authors · 2018-11-27
> **Suggested Densenet and adversarial experiments corroborate findings; wall clock was included in submission**
>
> We thank the reviewer for the detailed comments. We are happy that the reviewer recognized the importance of the problem and the potential relevance of the proposed solution across CNNs.
>
> We have updated the draft, with additional requested experiments in the appendix. We address all major and minor concerns below. In particular, we perform the requested experiments (DenseNet and adversarial attacks), which further corroborate findings in the submission.
>
> ----- TIMING -----
> > “Wall-clock times need to be reported for the various blurring kernels and compared to the baselines.”
>
> We agree that timing is an important consideration. In the submission, wall-clock times were reported in Table 2 and discussed in the last paragraph in Section 4. In summary, for VGG, the largest filter (7x7) adds 12.3% computation.
>
> > “Despite being more expensive, do dilations fix the issue of missing translation equivariance provably and not just approximately”
>
> Yes, removing strides and adding dilations, as we described in the end of Section 2, would preserve shift-equivariance. However, this costs immense computation, as each layer needs to be evaluated more densely. For the VGG network, this adds 4x, 16x, 64x, and 256x computation for conv2-conv5 layers, respectively. We added discussion to this point based on your suggestion.
>
> ----- REQUESTED EXPERIMENTS -----
> > “Extend results to a cutting-edge architecture, e.g. DenseNets or Wide ResNets.”
>
> Thank you for the suggestion. In Appendix A and Figure 8, we show the results applied to a more modern DenseNet (Huang et al., 2017) architecture. The results confirm the findings from the VGG architecture. In short:
> - When training without data augmentation, the proposed technique improves both classification consistency and accuracy over the baseline (as before).
> - The proposed technique trained without data augmentation outperforms the baseline, even with data augmentation.
> - When training with data augmentation, the proposed technique improves consistency, and surprisingly, even slightly improves accuracy in this setting.
>
> These results help support the general applicability of the method to CNNs.
>
> > “the authors should attack their augmented network with the translation attack of [Engstrom et al. In Arxiv, 2017.]”
>
> Thank you for the suggestion. In the submission, we show that classification accuracy is maintained, while consistency is improved. We thus expect the method to be robust to a shift-based adversary.
>
> In Appendix B and Figure 9, we confirm this hypothesis empirically. We compute classification accuracy, as a function of maximum adversarial shift. A max shift of 2 means the adversary can choose any of the 25 positions within a 5x5 window. For the classifier to “win”, it must classify all positions correctly. More detailed discussion is in Appendix B. In short:
> - The baseline is very sensitive to the adversary. Our proposed method dramatically decreases sensitivity to the adversary.
> - Again, our proposed method (with Binomial-7 filter) without augmentation is more robust than the baseline, trained with data augmentation.
>
> These results corroborate the findings in the main paper, and demonstrate a use case: increased robustness to shift-based adversarial attack.
>
> ----- WRITING -----
> Regarding the writing, we made minor edits in the main paper to the updated draft. We are happy the paper’s main ideas were “easy to follow”, and kept the overall structure. Based on your suggestion, we reduced the caption lengths. We are continuing to improve the paper.
>
> > “(Minor) Strange notation e.g. in equation 1. Why not write: x+\delta x in the argument of the function instead of "Shift". The current notation seems unnecessarily informal.”
>
> The Shift function is defined in Equation 4. Defining a shift function once enables reuse in six other locations (rather than using h-\delta h, w-\delta w indexing repeatedly), so we are inclined to keep it for now. Based on your comment, we added a note before Eqn 1, to better orient the reader.
>
> > “Figure 4: show scale and color bar.”
>
> Thank you for the suggestion. We will add a colorbar in an updated version and are continuing to improve the paper.

---

> > ### Comment · AnonReviewer1 · 2018-12-04
> > **Thank you very much!**
> >
> > The robustness is encouraging and the DesNet results substantiate the claims as well.
> > The updated manuscript is also easier to read now.
> >
> > I increased my score.
> >
> > To be clear, for a 7 or higher, I would need to see large scale experiments on multiple architectures and datasets. If the paper is rejected I encourage the authors to apply their proposed approach to ILSVRC2012/Cifar10/Cifar100 with VGG, Wide ResNet and DenseNet. ImageNet training is feasible in a few GPU days and for such a general claim, strong and broad empirical evidence is needed to be truly convincing.

---

> > > ### Author Response · Authors · 2018-12-05
> > > **Thank you, but we are somewhat confused**
> > >
> > > Thank you for reading the rebuttal. While we are happy the reviewer is open to making an adjustment, we admit to being rather perplexed by the additional, previously unmentioned, requirement for ImageNet, which is keeping the reviewer from adjusting the score beyond “marginally above”.
> > >
> > > The original review (“marginally below”) made the erroneous implication that wall clock times were not reported. We corrected this misconception in the rebuttal (it was in Table 2 of the original submission). Additionally, we performed all requested experiments -- adversarial attack and densenet -- which corroborated experimental results in the submission.
> > >
> > > This additional ImageNet request was not in the original review. While we are happy to conduct it, we are out of time. As our method is derived from “textbook” first-principles [1-4], we can reasonably expect empirical experiments to continue supporting the conclusions in the submission (Section 4.2) and rebuttal period (Appendices A-D).
> > >
> > > [1] Section 4.6.1: Sampling Reduction by an Integer Factor. Oppenheim, Schafer, Buck. Discrete-Time Signal Processing. 2nd ed. 1999
> > > [2] Section 2.4.5: Zooming and Shrinking Digital Images. Gonzalez and Woods. Digital Image Processing. 2nd ed. 1992.
> > > [3] Section 14.10.6: Antialiasing in Practice. Foley, van Dam, Feiner, Hughes. Computer Graphics: Principles and Practice. 2nd ed. 1995.
> > > [4] Section 3.5.2: Decimation. Szeliski. Computer Vision: Algorithms and Applications. 2010.

---

> > > > ### Comment · AnonReviewer1 · 2018-12-05
> > > > **Clarification**
> > > >
> > > > Sorry for making it sound that way, ImageNet is not a requirement, I did change my rating from weak reject to weak accept after all. I did this because of the additional experiments. Everything else I wrote are simply suggestions for possible future re-submissions.

---

### Public Comment · (anonymous) · 2018-10-31
**What is the difference between your blurring and a convolutional layer?**

I like the idea of this paper and I like to test the BlurDownsample for other purposes like for bottleneck autoencoders. I was wondering if the authors could prepare an implementation of the proposed BlurDownsample layer into one of the existing frameworks, e.g. Tensorflow.

One Question:
In Table 1, you evaluated 18 different filters and report that the Filter = [1,2,3,2,1] and the Filter = [1,6, 15, 20, 15, 6, 1] outcome the best results. So, it seems to me that you had a search on the different combinations for the Blurring filter to find a filter that gives you the best Test accuracy. So, is this the right technique or you should find the best filter through the cross-validation on training dataset?
And more importantly,  what if I use a convolutional layer which keep the dimensions of the input data same, and let the model finds the best filter through backpropagation, instead of using your Blurring layer with a fixed filter map? I mean, it seems to me your proposed layer is just a specific type of a convolutional layer, with a fixed filter map.

Some minor comments:
* "An L-layer deep can be..." --> "An -L-layers deep neural network can be.."
* "Modern convolutional networks are not shift-invariant...".  This is not a proper beginning for the abstract.  First, what do you mean by "Modern"?. Second, they are actually partially shift-invariant (Modulo-N as you shown in figure 2), and not fully shift-invariant.
* This paper deserves a more relevant title.

---

### Meta-Review · Area_Chair1 · 2018-12-14
**Anti-aliasing has been explored before.**

**Confidence:** 5
**Recommendation:** Reject

**Metareview:**

The reviewers are reasonably positive about this submission although two of them feel the paper is below acceptance threshold. AR1 advocates large scale experiments on ILSVRC2012/Cifar10/Cifar100 and so on. AR3 would like to see more comparisons to similar works and feels that the idea is not that significant. AR2 finds evaluations flawed. On balance, the reviewers find numerous flaws in experimentation that need to be improved.

Additionally, AC is aware that approaches such as 'Convolutional Kernel Networks' by J. Mairal et al. derive a pooling layer which, by its motivation and design, obeys the sampling theorem to attain anti-aliasing. Essentially, for pooling, they obtain a convolution of feature maps with an appropriate Gaussian prior to sampling. Thus, on balance, the idea proposed in this ICLR submission may sound novel but it is not. Ideas such as 'blurring before downsampling' or 'low-pass filter kernels' applied here are simply special cases of anti-aliasing. The authors may also want to read about aliasing in 'Invariance, Stability, and Complexity of Deep Convolutional Representations' to see how to prevent aliasing. On balance, the theory behind this problem is mostly solved even if standard networks overlook this mechanism. Note also that there exist a fundamental trade-off between shift-invariance plus anti-aliasing (stability) and performance; this being a reason why max-pooling is still preferred over anti-aliasing (better performance versus stability). Though, this is nothing new for those who delve into more theoretical papers on CNNs: this is an invite for the authors to go thoroughly first through the relevant literature/numerous prior works on this topic.

---

> ### Author Response · Authors · 2019-04-27
> **Updated paper accepted to ICML 2019**
>
> An updated paper has been accepted to ICML 2019. The project and paper is here: https://richzhang.github.io/antialiased-cnns/.
>
> The core of the paper remains the same, but with a major new result on Imagenet classification. We show improvements in both shift-invariance (as expected) and accuracy (surprisingly!), across popular networks - Alexnet, VGG, Resnet, and DenseNet. This result is contrary to popular belief, as expressed in the metareview: "Note also that there exist a fundamental trade-off between shift-invariance plus anti-aliasing (stability) and performance; this being a reason why max-pooling is still preferred over anti-aliasing (better performance versus stability)."
>
> I thank the AC for the pointer to the previous work, and address differences in the related work section, copied here for reference: "Mairal et al. (2014) derive a network architecture, motivated by translation invariance, named Convolutional Kernel Networks. While theoretically interesting (Bietti & Mairal, 2017), CKNs perform at lower accuracy than contemporaries, resulting in limited usage. Interestingly, a byproduct of the derivation is a standard Gaussian filter; however, no guidance is provided on its proper integration with existing network components. Instead, we demonstrate practical integration with any strided layer, and empirically show performance increases on a challenging benchmark -- ImageNet classification – on widely-used networks."
>
> Regarding novelty, we agree that anti-aliasing and low-pass filtering is obviously not novel. In the updated paper, we begin by saying, "When downsampling a signal, such an image, the textbook solution is to anti-alias by low-pass filtering the signal (Oppenheim et al., 1999; Gonzalez & Woods, 1992)." Our paper's contribution is to demonstrate harmonious integration of low-pass filtering with existing network components.
>
> Finally, I thank the ICLR metareviewer and reviewers. I greatly appreciate their informative comments and feedback through the review process!